# Age at Onset and Presenting Symptoms of Neurofibromatosis Type 2 as Prognostic Factors for Clinical Course of Vestibular Schwannomas

**DOI:** 10.3390/cancers12092355

**Published:** 2020-08-20

**Authors:** Isabel Gugel, Florian Grimm, Julian Zipfel, Christian Teuber, Ulrike Ernemann, Lan Kluwe, Marcos Tatagiba, Victor-Felix Mautner, Martin Ulrich Schuhmann

**Affiliations:** 1Department of Neurosurgery, University Hospital Tübingen, 72076 Tübingen, Germany; florian.grimm@med.uni-tuebingen.de (F.G.); julian.zipfel@med.uni-tuebingen.de (J.Z.); teuber_christian@web.de (C.T.); marcos.tatagiba@med.uni-tuebingen.de (M.T.); martin.schuhmann@med.uni-tuebingen.de (M.U.S.); 2Centre of Neurofibromatosis and Rare Diseases, University Hospital Tübingen, 72076 Tübingen, Germany; v.mautner@uke.de; 3Division of Pediatric Neurosurgery, University Hospital Tübingen, 72076 Tübingen, Germany; 4Department of Neuroradiology, University Hospital Tübingen, 72076 Tübingen, Germany; ulrike.ernemann@med.uni-tuebingen.de; 5Department of Neurology, University Medical Center Hamburg-Eppendorf, 20251 Hamburg, Germany; kluwe@uke.de; 6Department of Maxillofacial Surgery, University Medical Center Hamburg-Eppendorf, 20251 Hamburg, Germany

**Keywords:** presenting symptom, hearing preservation, neurofibromatosis type 2, growth rate, vestibular schwannoma

## Abstract

The presenting symptoms of the tumor suppressor gene syndrome neurofibromatosis type 2 (NF2) are often non-specific and unrelated to the disease hallmark bilateral vestibular schwannomas (VS). However, age at onset and presenting symptoms may have predictive values for the clinical course of VS. In this retrospective single-center study, we addressed this issue by reviewing 106 patients with 194 VS. Presenting symptoms attributable to VS commonly occur in 87% of adults and 31% of children. Age at onset significantly correlates with tumor volumes at presentation (*p* = 0.034). In addition, age at onset significantly correlates with pure-tone average (*p* = 0.0001), speech discrimination scores (*p* = 0.001), age at beginning of hearing loss (*p* = 0.0001), age at deafness (*p* = 0.0001), and age at first surgery (*p* = 0.0001). Patients presenting with VS related symptoms had significantly (*p* < 0.05) worse hearing values at presentation and after surgery. These patients also exhibited higher growth rates and tumor volumes compared to patients with non-VS related presenting symptoms, but this difference did not reach the significance level of *p* < 0.05. Due to the late appearance of these symptoms, the time of beginning hearing loss, surgery and deafness is significantly delayed (*p* < 0.05) compared to patients not presenting with VS. In summary, age at onset and type of presenting symptom provide excellent prognostic parameters for predicting VS- and hearing-related clinical course.

## 1. Introduction

Neurofibromatosis type 2 (NF2) is an autosomal-dominant tumor-prone disorder and caused by biallelic inactivation of the *NF2* tumor-suppressor gene on the chromosome band 22q12 [1,2]. Its typical hallmark is the presence of bilateral vestibular schwannoma (VS), which carries the risk of bilateral hearing loss and deafness over the lifetime [3,4]. Treatment options for these tumors are limited to microsurgery [5], chemotherapy, mainly with bevacizumab [6,7,8,9], and radiation [10]—all pursuing the major goal of tumor growth control and hearing preservation. Therefore, early diagnosis and the identification of prognostic parameters long before hearing deterioration occurs seem crucial in achieving this goal.

However, this is difficult due to the lack of specificity of VS unrelated presenting symptoms [11,12,13,14,15,16], such as ophthalmological findings (e.g., cataracts, retinal hamartomas, optic nerve sheath tumors, fibrotic maculopathies, and epiretinal membranes) [16,17,18,19,20], or cutaneous lesions (e.g., cutaneous schwannoma) [16,19,20], as well as muscle wasting due to neuropathy [21]. Although typical, and already included in the diagnostic NF2 criteria, they are often misinterpreted or not recognized as NF2 features. Other rare and non-NF2 specific events, such as vascular disease in form of brainstem strokes and aneurysm [22] lead to a non-delayed diagnostic assessment due to their dynamic nature and usually associated severe focal neurological deficits, which prompt a cranial magnetic resonance imaging (cMRI) that visualize asymptomatic VS establishing the diagnosis of NF2. Compared to children, adolescents and adults usually become symptomatic with VS related symptoms (hearing loss, tinnitus, facial palsy, gait disturbances) [4,21].

This study investigated the correlation of age and presenting symptoms at onset with the later clinical course of bilateral VS. The identification of prognostic parameters is valuable for the management of NF2 patients of all age groups.

## 2. Results

### 2.1. Patients, Tumors, and Presenting Symptoms

Detailed descriptive clinical and demographic data are summarized in Table 1, Table 2 and Table 3. Data of the individuals are given in the Appendix A.

The interval from symptom onset until the time of diagnosis was approximately 2 years in young NF2 patients (≤18 years (y) at the time of diagnosis) and 4 y in the adult group (>18 y). In both age groups, the average time from symptom onset to the beginning of hearing loss was ~9 y and to deafness further ~10 y.

Overall, 70% of tumors received treatment over time. Among the 121 operated tumors, 26 tumors were operated twice and one tumor three times due to continued tumor growth, large volumes and further hearing deterioration. For 14 ears no hearing data was available, 136 ears maintained their hearing during the investigated period (2004 until 2019), 44 ears became deaf either due to the natural (non-treated) course (50%), directly after surgery (43%) or during the time of bevacizumab treatment (7%). Presenting symptoms and pathologies in both age groups (≤18 and >18 y at time of diagnosis) are listed in Table 2 and Table 3.

VS related symptoms were seen in most of the patients (55%), followed by ophthalmological symptoms/abnormalities (45%) and cutaneous features (38%). About 40% of all patients were monosymptomatic, 55% of patients showed ≥2 symptoms at symptom onset and ~5% were asymptomatic. Including polysymptomatic occurrence and symptomatic as well as asymptomatic findings, 62% of young patients exhibit ophthalmological abnormalities, followed by cutaneous findings (43%).

In the vast majority of symptomatic cases (*n* = 75, 75%), the presenting symptoms corresponded to the symptoms leading to diagnosis. In the other 16 (≤18 y) and 9 (>18 y) patients presenting symptoms were not equal to those initializing the diagnostic process. Examinations resulting in the diagnosis of NF2 were initialized later upon newly developed symptoms, which were attributed to VS in most cases (*n* = 16, 64%) with hypacusis as major symptom (*n* = 12/16), followed by peripheral nerve tumors (*n* = 4, 16%), stroke (*n* = 2, 8%), spinal (*n* = 2), and intracranial non-VS related tumors (*n* = 1), cataract (*n* = 1), and neuropathy (*n* = 1).

### 2.2. Influence of Presenting Symptom(s) on Time to Diagnosis

The median values for “age at diagnosis in years” (*H* (5) = 33.139, *p* = 0.0005), “age at onset in years” (*H* (5) = 33.630, *p* = 0.0005), and “latency to diagnosis in months” (*H* (4) = 11.989, *p* = 0.017) were statistically significantly different between groups.

Subsequently, pairwise comparisons were performed using Dunn’s (1964) procedure with a Bonferroni correction for multiple comparisons. This post hoc analysis revealed statistically significant group differences which are illustrated and highlighted in Figure 1 (* adjusted *p*-values). All other comparisons showed no statistically significant difference in the mentioned categories.

### 2.3. Association of Age at Onset with VS Related Parameters

The multiple regression model statistically significantly predicted “Age at onset”, F (4, 142) = 3.254, *p* = 0.014, adj. R^2^ = 0.058. No significant correlation was seen between age at onset and growth rate at presentation, but growth rates at presentations seem to increase with increasing age.

All of the other six variables added statistically significantly to the prediction, *p* < 0.05. The correlation of parameters with age at onset is summarized in Table 4. There was a statistically significant, strong positive correlation between age at onset and age at deafness *r* (41) = 0.814, *p* < 0.0005, age at time of surgery *r* (117) = 0.835, *p* < 0.0005 and age at beginning of hearing loss *r* (131) = 0.816, *p* < 0.0005.

### 2.4. Association of Presenting Symptoms with VS Related Parameters

Distributions of the pure-tone (PTA), speech discrimination score (SDS), volume and growth rate scores, age at deafness, age at time of surgery, age at beginning of hearing loss were similar in the groups “VS presenting” and “non-VS presenting”, as assessed by visual inspection. Median scores for PTA and SDS were statistically significantly (*p* < 0.05) different between “VS presenting” and “non-VS presenting”. Although the significant level was not reached (*p* > 0.05), a tendency was seen towards larger volumes, increased growth rates at presentation, decreased growth rates after surgery as well as a later time of deafness in patients presenting with VS related symptoms. Growth rates under bevacizumab were almost similar in both groups. Detailed values are outlined in Table 5.

## 3. Discussion

Due to the bilateral presentation of vestibular schwannomas (VS) in patients with Neurofibromatosis type 2 (NF2), total and bilateral deafness is an expectable threat over lifetime, which can result in serious social and educational problems. Therefore, any treatment modality should have as its primary goals the delay of hearing loss and the preservation of the existing hearing as long as possible. To pursue these goals, early confirmation of diagnosis with detection of VS and monitoring of hearing is crucial.

The mechanism leading to hearing loss in both sporadic and NF2-associated VS is still not fully understood. It is assumed that the underlying pathophysiology is complex and several factors might be involved. These include mechanical effects leading to direct compression of adjacent nerval and neurovascular structures [23,24], as well as tumor secreting factors that cause a significant loss of hair cells [25] and elevated levels of perilymphatic proteins inducing a cochlear dysfunction [26]. These hypotheses can be supported by certain positive treatment responses such as the bony decompression of the internal auditory canal to relieve the structures [5,27] and systemic treatment with bevacizumab (reduction of tumor edema, tumor shrinkage [8] and possible restoration of protein homeostasis [26]).

Available and well-investigated treatment options are limited to surgery with variable degrees of resection extents [5,27], long-term systemic treatment with bevacizumab [9,28,29] and radiation [30] but the responses are heterogeneous and can be associated with side-effects and complications. Radical total tumor removal carries a high risk of surgically induced deafness as we could show that hearing preservation is dependent on the extent of resection [27].

The main success factors for long-term hearing preservation are early detection and frequent hearing monitoring in these tumors, which enables to initiate treatment when hearing begins to be affected but is not yet functionally impaired. In terms of surgery, this means the better the hearing before, the better after surgery [27]. In the presented data we could demonstrate, that patients becoming symptomatic early—mostly with non-VS specific symptoms, such as ophthalmological or cutaneous lesions—had better hearing values, smaller tumor volumes, and a tendency towards lower growth rates than those patients with a later onset. However, on the other hand, the beginning of hearing loss is earlier and thus these patients are at risk of becoming deaf and requiring surgery earlier as well.

In general, early age at onset is an important prognostic factor for patients with NF2, since earlier onset exhibits a poor prognosis and severe clinical course [19,21,31,32]. This is also due to the known genotype-phenotype correlations which are mainly based on the onset of disease [33]. For instance patients with a Wishart Type characterized by an early onset and a high tumor burden exhibit a younger age at death [21].

Despite age at onset, VS itself seems to be a risk factor for clinical and treatment outcomes. While adults commonly present with VS associated symptoms (87% in our data) [4,21], children and young adolescence usually exhibit non-VS related symptoms [11,13,14,18,20,34,35,36]. As patients presenting with symptoms attributable to VS show worse hearing values at presentation and after surgery and a tendency towards larger tumor volumes and growth rates, the recognition of the non-VS related—often prevenient presenting symptoms—is important and offers a chance. This is especially important in pediatric NF2 patients as they come into medical attention with ophthalmological, cutaneous or non-VS related neurological symptoms, which are often misinterpreted [20] and, thereby, a potentially avoidable delay in diagnosis occurs [17,18,19]. Approximately 50% of pediatric NF2 patients exhibit these symptoms, which occur on average between the ages of 9 and 12 years, according to our data but can already be present from birth on. Of course, this must be set into relation to the general incidence/prevalence for the common ophthalmological findings such as childhood cataracts (incidence of 1.8 to 3.6 per 10,000 [37]) and childhood strabismus (prevalence of 1 to 6%) [38]. Certainly, there is a non-negligible majority of non-NF2 specific causes in the population for these ophthalmological findings, but in differential diagnosis, rare causes, such as NF2, have to be considered at an early stage as well. In addition, the distinction of cutaneous lesions (e.g., café au lait spots or skin tumors) to other neuro-cutaneous syndromes such as Neurofibromatosis type 1 is sometimes challenging [21,39,40,41]. It should also be taken into account that an absence of bilateral VS in MR imaging in young children, while other symptoms are already present, is no exclusion of NF2, as these tumors often are not yet developed in very young children and MR imaging has to be repeated in the following years.

We advocate that all children and young adolescents manifesting with the herein presented features (also described in detail in the study of Gugel et al. [20]) should be referred to a specialized NF care center. Particularly young patients presenting with isolated schwannomas or meningiomas should undergo early genetic testing for tumor predisposition syndromes [12,42,43], and some recommendations for such cases are illustrated in the study of Pathmanaban et al. [42].

Certainly, due to the retrospective character of this single-center analysis, the presented data should be treated with some caution, cannot be generalized, and used for predictive values. Multi-center prospective analyses with larger patient numbers would be desirable but are difficult to realize due to the rarity of the disease, small case numbers, and the necessary long-term data. A clustering of NF referral beyond the national borders is necessary.

However, even in a multicenter analysis, reproducibility and uniformity despite guidelines on performance can be problematic, especially in the case of non-objective and non-instrumental evaluation for instance in ophthalmological examinations, which require a high degree of comparable expertise of the single investigator.

Despite the often-described low evidence level (Level 3) of retrospective case series, they are especially important for rare diseases as they can provide features leading to a full characterization, diagnosis and treatments and therefore improve the management and the understanding of rare diseases [44,45].

## 4. Material and Methods

### 4.1. Patients and Clinical Characteristics

A total of 106 NF2 patients (61 patients ≤18 years and 45 patients >18 years at the time of diagnosis) and with bilateral vestibular schwannomas were included in this single-center retrospective analysis between 2004 and 2019. A total of 18 of initially 212 tumors were excluded due to the lack of data and thus 194 Vs remained for the analysis.

The Ethics Board of the Medical Faculty and the University Hospital of Tübingen approved this retrospective analysis (No 018/2019BO2). All patients fulfilled the Baser diagnostic criteria [46] for NF2 and were screened by cranial and spinal imaging, ophthalmologic and neurologic examination. In 121 tumors, surgery in form of decompression of the internal auditory canal (IAC) with various resection amounts guided by electrophysiology was performed via a retrosigmoid approach. Further 59 patients were treated with bevacizumab either as the only treatment or as adjunct pre- or post-operatively. Six tumors were irradiated in the course of the disease. Indications for surgical treatment were (a) large tumors (T4 Hannover Classification [47]) on both sides with brainstem compression or (b) continuing tumor growth and deterioration of auditory evoked potential or impairment of pure-tone average (PTA) or speech discrimination score (SDS), during observation. In case of further tumor growth combined with hearing deterioration, patient wish or in case of difficult surgical conditions, treatment with bevacizumab was performed. In most cases, this was applied as a second-line treatment rather than first-line therapy, particularly in children and adolescence. Irradiation was reserved for some cases of very small tumors that do not fill out properly the IAC.

Clinical data were obtained from clinical reports and magnetic resonance imaging (MRI). Tumor volumetry, growth rates, and resection amounts were assessed, classified, and calculated in ~2500 data sets, as previously described [5]. NF2 genetic testing results were available for 69 patients. Moreover, 13 patients were diagnosed to be mosaic. No mutation could be found in either blood or tumor DNA in 7 patients.

Hearing was assessed in all patients by regular determination of 4-frequency PTA and SDS, within 4 weeks before surgery and then every three-to-six month after surgery/irradiation or during bevacizumab treatment. Inclusion criteria were the availability of relevant data (age/clinical presentation at diagnosis and symptom onset as well as tumor volume at first cranial MRI).

### 4.2. Statistical Analysis

For the statistical analysis, age at onset and VS related presenting symptoms were defined as dependent variables and growth rate (in cm^3^/year), volume (cm^3^), PTA (dB), and SDS (%) at presentation as well as age at time of deafness, at time of surgery and beginning hearing loss (all in years) as independent variables.

Statistical analysis was performed in SPSS (IBM SPSS Statistic for Windows, Version 22.0., IBM Corp, Armonk, NY, USA). Multiple regression and Pearson’s correlation coefficient were assessed to predict age at onset from the independent variables. Two outliers had to be excluded (ID 17.2 and 18.2) due to the wide range of data. There was linearity as assessed by partial regression plots and a plot of studentized residuals against the predicted values. There was independence of residuals, as assessed by a Durbin-Watson statistic of 0.802. There was homoscedasticity, as assessed by visual inspection of a plot of studentized residuals versus unstandardized predicted values. There was no evidence of multicollinearity, as assessed by tolerance values greater than 0.1. There were no studentized deleted residuals greater than ±3 standard deviations, no leverage values greater than 0.2, and values for Cook’s distance above 1. The assumption of normality was met, as assessed by a Q-Q Plot.

A Kruskal-Wallis H test was run to determine if there were differences in “age at diagnosis in years”, “age at onset in years” and “latency to diagnosis in months” between five categories of first symptom(s) “ophthalmological” (comprising cataracts, retinal hamartoma, strabismus, proptosis, visual impairment or loss, *n* = 48, “cutaneous/subcutaneous features” (comprising cutaneous (plexiform), and subcutaneous schwannoma and café au lait spots, *n* = 40), “VS” (symptoms due to VS such as hearing loss, sudden hearing loss, tinnitus, gait disturbances, facial palsy, *n* = 58), “non-VS related neurological symptoms” (comprising sensory and motor deficits, fine motor skill disorders, non-VS related facial palsy, dysarthria, memory impairment, seizures, muscle atrophy, vomiting/nausea, *n* = 31), and “pain” (e.g., headache, radiating pain, back and neck pain, *n* = 11). The conducted Kruskal-Wallis test revealed similar distributions of “age at diagnosis in years”, “age at onset of first sign(s)/symptom(s) in years” and “latency to diagnosis in months” for all groups as assessed by visual inspection of a boxplot.

A Mann-Whitney U and Wilcoxon signed-ranked test was run to determine if there were differences in the independent variables between patients/tumors presenting (“VS presenting symptom”) or not presenting (“non-VS presenting symptom”) with symptoms attributable to VS.

## 5. Conclusions

In children and adolescence who present with ophthalmological and cutaneous findings, as well as with solitary intracranial, spinal, or peripheral tumor lesions or vascular disease, the underlying diagnosis of NF2 should always be considered and clarified sooner rather than later. Early age at onset/diagnosis offers the chance of early monitoring in a usually good condition of hearing and thus allows early therapeutic intervention at a beginning hearing loss. However, early age at onset and presenting symptoms due to VS represent significant risk factors for poor outcome of these tumors.

## Figures and Tables

**Figure 1 cancers-12-02355-f001:**
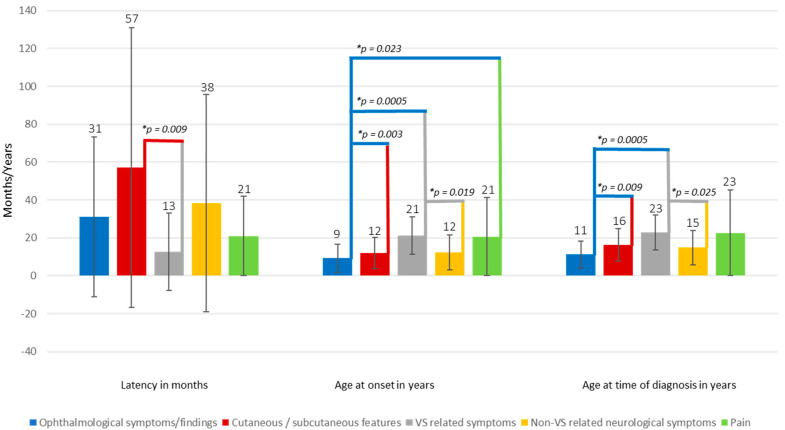
Age distribution of presenting symptom(s) and latency until time of diagnosis. Error bars represent the standard deviation.

**Table 1 cancers-12-02355-t001:** Demographic data of 106 neurofibromatosis type 2 (NF2) patients (194 tumors).

Age at Time of Diagnosis in Years	All Age Groups	≤18	>18
No. of patients/tumors	106/194	61/115	45/79
Sex (No. of female/male)	55/51	31/30	24/21
Age at in year (mean ± SD, range)			
Onset	16 ± 11, 0–46	9 ± 6, 0–18	25 ± 9, 0–46
Diagnosis	18 ± 10, 1–44	11 ± 5, 1–18	29 ± 7, 19–44
Deafness	26 ± 11, 13–59, *n* = 44	19 ± 5, 13–31, *n* = 24	
			35 ± 11, 22–59,
			*n* = 20
Time of surgery per ear	24 ± 10, 9–56, *n* = 121	18 ± 5, 9–30, *n* = 73	33 ± 8, 22–56, *n* = 48
At beginning of hearing loss	24 ± 11, 7–58, *n* = 135	18 ± 5, 7–30, *n* = 81	34 ± 9, 19–58, *n* = 54
Family history of NF2 - yes/no	21/85	12/49	9/36
Detected mutation types	In 49 patients	In 41 patients	In 8 patients
Splicing mutations	11	8	3
Nonsense mutations	11	8	3
Frameshifting mutations	14	14	0
Large genome alteration	1	1	0
Missense mutation	2	0	2
Large deletion	5	5	0
Mosaic	13 patients	2 patients	11 patients
No mutation detected in blood and tumor DNA	7 patients	3 patients	4 patients

At presentation (mean ± SD, range)			
Tumor volume in cm^3^			
	2.86 ± 5.73, 0–32.72,	2.39 ± 5.61, 0–33,	3.55 ± 5.83, 0.02–28.05,
	*n* = 194	*n* = 115	*n* = 79
Growth rate in cm^3^/year			
	0.82 ± 1.95, −1.66–14.96,	0.72 ± 1.71, −1.66–10.84,	0.96 ± 2.24, −0.08–14.96,
	*n* = 159	*n* = 93	*n* = 66
PTA in dB			
	31.22 ± 37.81, 1–130,	21.45 ± 29.4, 1.25–130,	44.91 ± 43.61, 1–130,
	*n* = 185	*n* = 108	*n* = 77
SDS in %			
	77.94 ± 35.85, 0–100,	86.93 ± 28.87, 0–100,	65.39 ± 40.56, 0–100,
	*n* = 183	*n* = 106	*n* = 77

No—Number; SD—standard deviation; y—years; PTA—pure-tone average; SDS—speech discrimination score.

**Table 2 cancers-12-02355-t002:** Details of presenting symptom(s) of 106 patients with NF2.

First Symptom(s)/Sign(s)	In All	≤18	>18
**Ophthalmological**	36	31	5
Visual impairment/loss *^a^	16	13	3
Strabismus *^b^	17	15	2
Proptosis *^c^	3	3	0
**VS related symptoms**	58	19	39
Hypacusis	26	9	17
Sudden hearing loss	5	0	5
Tinnitus	10	1	9
Balance disorders	5	3	2
Dizziness	5	1	4
Hoarseness	1	1	0
Facial palsy (due to VS)	6	4	2
**Non-VS related neurological symptoms**	31	18	19
Seizures	5	3	2
Motor deficits	9	7	2
Sensory deficits	2	2	0
Dysarthria	2	1	1
Fine motor skill disorder	1	0	1
Facial palsy (non-VS related)	4	2	2
Muscle wasting *^d^	5	2	3
Memory impairment *^e^	1	1	0
Vomiting/nausea *^f^	2	0	2
**Pain**	11	4	7
Back-/neck-/and radiating pain	9	4	5
Headache	2	0	2
**Family history/incidental finding**	10	5	5
Positive family history	5	3	2
Incidental finding (asymptomatic)	6	2	4
**Mono-/Polysymptomatic**			
Monosymptomatic patients	42	23	19
Polysymptomatic patients (≥2 symptoms)	58	35	23
Asymptomatic	6	3	3

Values are numbers of patients. VS—vestibular schwannoma. *^a^ Visual impairment was tumor-associated in 3 patients due to optic nerve sheath meningiomas and idiopathic in 5. In the remaining 8 patients, the visual impairment was attributed to cataracts or retinal hamartomas/maculopathy. *^b^ Strabismus was idiopathic in 13 cases and tumor-associated in 4 cases. *^c^ Patients with proptosis due to intraorbital tumor manifestations had no visual impairment. *^d^ Neuropathy was the cause of muscle wasting. *^e^ Hydrocephalus was the etiology for memory impairment. *^f^ Intracranial pressure symptoms vomiting and nausea were related to an intracranial meningioma and hydrocephalus due to a large VS.

**Table 3 cancers-12-02355-t003:** Causal pathology for presenting symptom(s) in 100 symptomatic NF2 patients.

Presenting Symptom (Related) Pathology/Feature	All	≤18	>18
Vestibular schwannoma (VS)	58(55%)	19(31%)	39(87%)
Cutaneous (plexiform)/subcutaneous schwannoma	37(35%)	25(41%)	12(27%)
Intracranial meningioma *^a^	12(11%)	7(11%)	5(11%)
Cataract	11(10%)	10(16%)	1(2%)
Retinal Hamartoma/Maculopathy	9(8%)	10(16%)	5(11%)
Intracranial non-VS schwannoma *^b^	6(6%)	4(7%)	2(4%)
Neuropathy/Muscle wasting	4(4%)	3(5%)	1(2%)
Café au lait spots	3(3%)	1(2%)	2(4%)
Ischemic brainstem stroke	2(2%)	1(2%)	1(2%)
Spinal extramedullary tumors (schwannoma, meningioma)	2(2%)	2(3%)	0
Spinal intramedullary tumor (ependymoma)	2(2%)	1(2%)	1(2%)
SAH bleeding due to aneurysm	1(1%)	1(2%)	0
Cerebral dysplasia	1(1%)	1(2%)	0
Bifrontal angiomatosis	1(1%)	1(2%)	0

Values are numbers of patients. VS—vestibular schwannoma; SAH—subarachnoid hemorrhage; 6/106 patients of the cohort were asymptomatic. *^a^ Intracranial meningioma were localized temporal, frontal, intraventricular craniocervical and at the cavernous sinus and optic nerve sheath as well as in the cerebellopontine angle. *^b^ Schwannomas originating from the III and VII cranial nerve and undefined schwannoma mass in the periorbital region.

**Table 4 cancers-12-02355-t004:** Correlations of VS related parameters with age at onset.

**Positive Correlation of Age at Onset with**	
GR at presentation in cm^3^/year	r = 0.065, *p* = 0.217
Volume at presentation in cm^3^	r = 0.150, *p* = 0.034
PTA at presentation in dB	r = 0.283, *p* = 0.0001
Age at deafness in years	r = 0.814, *p* = 0.0001
Age at time of surgery in years	r = 0.835, *p* = 0.0001
Age at beginning of hearing loss in years	r = 0.816, *p* = 0.0001
**Negative (Inversed) Correlation of Age at onset with**	
SDS at diagnosis in %	r = −0.252, *p* = 0.001

Note. VS—vestibular schwannoma; PTA—pure-tone average; SDS—speech discrimination score; GR—growth rate; dB—decibel.

**Table 5 cancers-12-02355-t005:** Difference in VS related parameters between patients with VS related and with non-VS related presenting symptoms.

Variable	*U*	z	*p*	Median Non-VS Presenting Symptom	Median VS Presenting Symptom
GR at presentation	2871	−0.469	0.639	0.138 cm^3^/year	0.169 cm^3^/year
Volume at presentation	3743	−1.758	0.079	0.389 cm^3^	0.905 cm^3^
PTA at presentation	3152	−2.727	**0.006**	10 dB	25 dB
SDS at presentation	3081	−2.852	**0.004**	100%	95%
Age at deafness	250	1.034	0.301	22 years	25 years
Age at time of surgery	2272	2.727	**0.006**	19 years	24 years
Age at beginning of hearing loss	2711	2.361	**0.018**	19 years	23 years
GR postoperative	1332	−1.745	0.081	0.276 cm^3^/year	0.069 cm^3^/year
PTA postoperative	1779	2.689	**0.007**	26 dB	55.25 dB
SDS postoperative	1006	−2.175	**0.03**	85%	50%
GR under BVZ	538	0.609	0.542	0.066 cm^3^/year	0.063 cm^3^/year

Note. Significant *p* values between the two groups are written in bold. PTA—pure-tone average; SDS—speech discrimination score; GR—growth rate; dB—decibel; VS—vestibular schwannoma; BVZ—bevacizumab.

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
