# Peer review of "Age at Onset and Presenting Symptoms of Neurofibromatosis Type 2 as Prognostic Factors for Clinical Course of Vestibular Schwannomas"

_cancers, 2020, doi:10.3390/cancers12092355_

Round 1

Reviewer 1 Report

The Authors try to present the result of a retrospective personal series of Neurofibromatosis 2 patients (NF2)  Theu report thet in childhood and adolescence NF2 onset is mostly characterized  by non-specific symptoms including between others, cataracts  retinal hamartomas, tumors of the optic sheats, fibrotic maculopathy or skin schwannomas, which are not helpful to address the diagnostic protocol towards studies useful to identify  NF2-associated  vestibular schwannomas.  Conversely  adult NF2 patients who presented with hypoacusis, tinnitus and sudden hearing loss were quickly diagnosed as NF2 paetients with associated VS. As in both group the  diagnosisi of VS is definite by CT/MRI studies , presentation and discussion  of neuro-imaging findings would be advaisable in this  series. The Autor, instead, limit thei comment to  the sentence that symptoms and signs  at onset   were likely misinterpreted or not recognized being suggestive of NF2 in childhood en adolescence.

 A huge mass of data is attached in which the reader cannot extricate himself, which is not discussed.

English is approximate, several sentences  lack  of verb. Terms like signs, symptoms, manifestations and findings  are  interchangeably used as synonyms. The meaningless sentence “signs of symptoms...” is repeated again and again.   I think this article cannot be considered for publication

Reviewer 2 Report

Overall comments

Case series of Schwannomas.  Case series have limited scientific value and should always be approached with care.  The study is well thought out and has specific aims that they are trying to answer with the data. The authors have spent a lot of time here…Could we ask for confidence intervals around %s?

Need to write in paragraphs.  Tables and text should be able to be read independently.

Specific comments

Table 2 seems to be raw data:  please supply analysis.

Table 3: I cant see any proportions

Figure 1: Y axis label?  Is in N?  is it the average age at time of diagnosis? If it is an average it needs error bars

Line 203: im not sure the rarity of the disease is brought across here.

Line 209: what is vs.?  is it VS?  choose one.

Reviewer 3 Report

In this manuscript, the authors looked at the prognostic factors associated with NF2 associated vestibular schwannomas. Nicely written, results make sense. I have a few comments at this point:

  • Is this is a single-center study? Please mention clearly in the appropriate places (methods etc)
  •  This is a retrospective study. Please mention limitations at the end of the discussion. Generalizability, single-center, retrospective, etc.
  • Please rewrite the conclusion section. It seems like the conclusion is derived from a literature review rather than the results of the manuscript. 

Let me know if you have any questions.

Thank you.
